# GRIFFIN: A C++ library for electroweak radiative corrections in fermion scattering and decay processes

Lisong Chen[1,2] and Ayres Freitas[1]

[1]Pittsburgh Particle-physics Astro-physics & Cosmology Center (PITT-PACC),, Department of Physics & Astronomy, University of Pittsburgh, Pittsburgh, PA 15260, USA
[2]Institut für Theoretische Teilchenphysik, Karlsruhe Institute of Technology (KIT), Wolfgang-Gaede Straße 1, 76128 Karlsruhe, Germany

## Abstract

This paper describes a modular framework for the description of electroweak scattering and decay processes, including but not limited to Z-resonance physics. The framework consistently combines a complex-pole expansion near an s-channel resonance with a regular fixed-order perturbative description away from the resonance in a manifestly gauge-invariant scheme. Leading vertex correction contributions are encapsulated in form factors that can be predicted or treated as numerical fit parameters.

This framework has been implemented in the publicly available object-oriented C++ library GRIFFIN. Version 1.0 of this library provides Standard Model predictions for the IR-subtracted matrix elements for the process $f\bar{f} \to f'\bar{f}'$ with full NNLO and leading higher-order contributions on the Z-resonance, and with NLO corrections off-resonance. The library can straightforwardly be extended to include higher-order corrections, should they become available, or predictions for new physics models. It can be interfaced with Monte-Carlo programs to account for QED and QCD initial-state and final-state radiation.

## 1   Introduction

Studies of fermion scattering, $f\bar{f} \to f'\bar{f}'$ for center-of-mass energies near the Z-boson resonance, $\sqrt{s} \sim M_Z$, have played a crucial role in elucidating the structure of the Standard Model (SM) and putting constraints in potential new physics beyond the SM (BSM). These include precision measurements at LEP and SLC (where $f = e$) [1], as well as Drell-Yan production at the TeVatron and LHC (where $f = u, d, s, c, b$) [2–4]. Even higher levels of precision can be achieved at the high-luminosity run of the LHC (HL-LHC) [5] and one of several proposed future $e^+e^-$ colliders: FCC-ee [6], CEPC [7], ILC [8,9], CLIC [10,11].

The relevant outcomes of these experiments are typically presented in terms of a set of so-called electroweak pseudo-observables (EWPOs) that encapsulate the dominant radiative

corrections in the SM and are most sensitive to BSM physics. Examples of EWPOs are effective Z-fermion couplings, partial Z-boson widths, the effective weak mixing angle $\sin^2 \theta_{\text{eff}}^f$, and the asymmetry parameters $A_f$; see *e.g.* Ref. [1] for the definition of these quantities. For the full description of the observable cross-sections, however, a number of other ingredients besides the EWPOs are needed, such as contributions from diagrams without s-channel Z-bosons (*i.e.* s-channel photon exchange and box diagrams) and initial- and final-state QED and QCD radiation effects. A number of software packages provide these ingredients with complete next-to-leading order (NLO) and some partial higher-order corrections included [12–29]. Among these, the ZFITTER [14,15] and TOPAZ0 [13] packages also provide extensive formulae for real photon radiation contributions, including certain selection cuts. They have been widely used in experimental studies. One can refer to [30] for the most recent updates on such analytical methods. Alternatively, QED radiation can be simulated with Monte-Carlo (MC) methods. For example, the electroweak corrections provided by the package DIZET [12], which is a component of ZFITTER, have been linked to the MC programs KoralZ [31] and KKMC [32][1]

However, despite the tremendous success of these software tools, they may not be easily adaptable to future applications that require a higher level of precision. Such applications call for a setup that enables the incorporation of higher-order corrections (NNLO and beyond) in a well-controlled and manifestly gauge-invariant way, as well as a modular object-oriented structure for the straightforward implementation of new SM or BSM contributions. In this article, the new software package GRIFFIN (Gauge-invariant Resonance In Four-Fermion INteractions) is introduced, which aims to provide a framework with these desirable features. It is written in C++ and defines a class hierarchy that can be extended with new results (both in the SM and beyond) without modifying its interface to external users (such as MC or fitting programs). While its object-oriented structure is, in principle, general enough to implement any arbitrary physics process, the current version is focused on 4-fermion processes, *i.e.* scattering processes of the form $f\bar{f} \to f'\bar{f}'$ or decay processes like muon decay. The relevant matrix elements for these processes are implemented in IR-subtracted form, which can be interfaced with the MC program to treat QED and QCD radiation. To describe the Z-boson resonance, it uses a Laurent expansion of the hard matrix elements[2] about the complex pole $s_0 \equiv M_Z^2 - iM_Z\Gamma_Z$. Since this pole is an analytical property of the S-matrix, both the pole location and the coefficients of the expansion are individually gauge-invariant [34–39].

The paper is organized as follows. Section 2 introduces the formalism for the complex-pole expansion and discusses what building blocks are required to describe the Z resonance at NNLO precision. On the other hand, outside of the resonance region, no pole expansion is needed. In section 3, it is discussed how on- and off-resonance predictions can be consistently matched to obtain reliable results for $f\bar{f} \to f'\bar{f}'$ at any center-of-mass energy. The implementation of these elements within the GRIFFIN library is described in section 4, and

---

[1]Similar functionality, without using the DIZET package, is also provided by a number of other MC tools. See Ref. [33] for a broader overview.

[2]Here "hard" refers to the matrix element without initial-state and final-state QED/QCD radiation since the former would produce a deformation of the resonance lineshape.

numerical results and comparisons with ZFITTER/DIZET are shown in section 5. Finally, a summary is provided in section 6.

## 2 Fermion pair production on the Z resonance

The matrix element for the process $f\bar{f} \to f'\bar{f}'$ can be decomposed into four different chirality structures, which here will be delineated according to their vector or axial-vector couplings:

$$\mathcal{M} = \left[ M_{\text{VV}} \gamma^\mu \otimes \gamma_\mu - M_{\text{VA}} \gamma^\mu \otimes \gamma_\mu \gamma^5 - M_{\text{AV}} \gamma^\mu \gamma^5 \otimes \gamma_\mu + M_{\text{AA}} \gamma^\mu \gamma^5 \otimes \gamma_\mu \gamma^5 \right], \qquad (1)$$

where the $\otimes$ stands for the outer product of two fermion chains. In terms of these quantities, the differential cross-section is given by

$$\frac{d\sigma}{d\cos\theta} = \frac{N_c}{32\pi s} |\mathcal{M}|^2 \qquad (2)$$

$$= \frac{N_c s}{32\pi} \left[ (1 + c_\theta^2)(|M_{\text{VV}}|^2 + |M_{\text{VA}}|^2 + |M_{\text{AV}}|^2 + |M_{\text{AA}}|^2) \right. \qquad (3)$$

$$+ 4c_\theta \operatorname{Re}\{M_{\text{VV}} M_{\text{AA}}^* + M_{\text{VA}} M_{\text{AV}}^*\}$$

$$- 2P_f(1 + c_\theta^2) \operatorname{Re}\{M_{\text{VV}} M_{\text{AV}}^* + M_{\text{VA}} M_{\text{AA}}^*\}$$

$$\left. - 4P_f c_\theta \operatorname{Re}\{M_{\text{VV}} M_{\text{VA}}^* + M_{\text{AV}} M_{\text{AA}}^*\} \right],$$

where $c_\theta = \cos\theta$, $\theta$ is the scattering angle in the center-of-mass frame, and $s$ is the center-of-mass energy, $P_f$ is the polarization of the incoming fermion $f$, and the masses of $f$ and $f'$ have been neglected. For Z-boson exchange at tree level, the four chiral matrix elements read

$$M_{\text{VV}}^{(0)} = \frac{v_{f(0)}^{\text{Z}} v_{f'(0)}^{\text{Z}}}{s - s_0}, \quad M_{\text{VA}}^{(0)} = \frac{v_{f(0)}^{\text{Z}} a_{f'(0)}^{\text{Z}}}{s - s_0}, \quad M_{\text{AV}}^{(0)} = \frac{a_{f(0)}^{\text{Z}} v_{f'(0)}^{\text{Z}}}{s - s_0}, \quad M_{\text{AA}}^{(0)} = \frac{a_{f(0)}^{\text{Z}} a_{f'(0)}^{\text{Z}}}{s - s_0}, \qquad (4)$$

where $s_0 \equiv M_{\text{Z}}^2 - iM_{\text{Z}}\Gamma_{\text{Z}}$ and

$$v_{f(0)}^{\text{Z}} = \frac{eI_f^3(1 - 4|Q_f|s_{\text{w}}^2)}{2s_{\text{w}}c_{\text{w}}}, \qquad a_{f(0)}^{\text{Z}} = \frac{eI_f^3}{2s_{\text{w}}c_{\text{w}}} \qquad (5)$$

are the vector and axial-vector couplings of the Z-boson to the fermion $f$ ($f = \ell, \nu, u, d, ...$). Furthermore, $s_{\text{w}}$ and $c_{\text{w}}$ stand for the sine and cosine of weak-mixing angle.

Note that throughout this document, $M_{\text{Z}}$ and $\Gamma_{\text{Z}}$ refer to the mass and width of the Z-boson in the complex-pole scheme, which is theoretically well-defined and gauge-invariant [34–39]. However, most experimental measurements are typically reported in terms of the so-called running-width scheme, leading to different values for the mass and width, which we denote as $M_{\text{Z}}^{exp}$ and $\Gamma_{\text{Z}}^{exp}$. The two definitions are related via [40][3]

$$M_{\text{Z}} = M_{\text{Z}}^{exp} (1 + (\Gamma_{\text{Z}}^{\text{exp}}/M_{\text{Z}}^{exp})^2)^{-1/2}, \qquad \Gamma_{\text{Z}} = \Gamma_{\text{Z}}^{\text{exp}} [1 + (\Gamma_{\text{Z}}^{\text{exp}}/M_{\text{Z}}^{exp})^2]^{-1/2}. \qquad (6)$$

---

[3]See Ref. [41] for a general review of the treatment of electroweak gauge-boson resonances.

In general, when including photon-exchange diagrams and higher-order contributions, the matrix elements can be written as Laurent expansion about the complex pole $s_0$,

$$M_{ij} = \frac{R_{ij}}{s - s_0} + S_{ij} + (s - s_0)S'_{ij} + \dots \qquad (i, j = \text{V, A}). \tag{7}$$

Note that the scattering angle $\theta$ is kept fixed when expanding $M_{ij}(s, \theta)$. To construct explicit expressions for $R, S, S'$, we introduce the following quantities:

$$Z_{Vf}(s) \equiv v_f^Z(s) + v_f^\gamma(s)\frac{\Sigma_{\gamma Z}(s)}{s + \Sigma_{\gamma\gamma}(s)}, \qquad\qquad G_{Vf}(s) \equiv v_f^\gamma(s), \tag{8}$$

$$Z_{Af}(s) \equiv a_f^Z(s) + a_f^\gamma(s)\frac{\Sigma_{\gamma Z}(s)}{s + \Sigma_{\gamma\gamma}(s)}, \qquad\qquad G_{Af}(s) \equiv a_f^\gamma(s), \tag{9}$$

$$\Sigma_Z(s) \equiv \Sigma_{ZZ}(s) - \frac{[\Sigma_{\gamma Z}(s)]^2}{s + \Sigma_{\gamma\gamma}(s)}. \tag{10}$$

Here $v_f^V$ ($a_f^V$) is the vector (axial-vector) form factor for the vertex between the gauge boson V (V=Z,$\gamma$) and the fermion $f$, including loop contributions. $\Sigma_{V_1 V_2}$ is the self-energy for incoming $V_1$ and outgoing $V_2$ ($V_{1,2}$=Z,$\gamma$). Furthermore, we denote

$B_{ij}(s, t)$ : Contribution of $\gamma\gamma$, $ZZ$ and $WW$ box diagrams for initial-state (11) vector/axial-vector current ($i = $ V, A) and final-state vector/axial-vector current ($j = $ V, A);

$$B_{\gamma Z, ij}(s, t) = \frac{B^R_{\gamma Z, ij}}{s - s_0} + B^S_{\gamma Z, ij} + (s - s_0)B^{S'}_{\gamma Z, ij} + \dots :$$

Contribution of $\gamma$Z box diagrams, which can also contribute to the (12) leading pole term $R_{ij}$.

It should be noted that the coefficients $B^{R,S,S',\dots}_{\gamma Z, ij}$ contain additional logarithms $\ln(1 - \frac{s}{s_0})$ that become singular on the pole and need to be accounted for in the Laurent expansion. Up to one-loop order, one thus has $B^X_{\gamma Z, ij} = B^{X,1}_{\gamma Z, ij}\ln(1 - \frac{s}{s_0}) + B^{X,0}_{\gamma Z, ij}$ ($X = R, S, S', \dots$), where $B^{X,i}_{\gamma Z, ij}$ are independent of $s$. In practice, these contributions are calculated by expanding the full analytical expression for the one-loop $\gamma$Z box diagrams while tracking both polynomial and logarithmic singularities.

In terms of the above quantities, the coefficients of the complex-pole expansion are read

$$R_{ij} = \left[\frac{Z_{if}Z_{jf'}}{1 + \Sigma'_Z}\right]_{s=s_0} + B^R_{\gamma Z, ij}, \tag{13}$$

$$S_{ij} = \left[\frac{Z_{if}Z'_{jf'} + Z'_{if}Z_{jf'}}{1 + \Sigma'_Z} - \frac{Z_{if}Z_{jf'}\Sigma''_Z}{2(1 + \Sigma'_Z)^2} + \frac{G_{if}G_{jf'}}{s + \Sigma_{\gamma\gamma}} + B_{ij}\right]_{s=s_0} + B^S_{\gamma Z, ij}, \tag{14}$$

$$S'_{ij} = \left[\frac{Z_{if}Z''_{jf'} + Z''_{if}Z_{jf'} + 2Z'_{if}Z'_{jf'}}{2(1 + \Sigma'_Z)} - \frac{(Z_{if}Z'_{jf'} + Z'_{if}Z_{jf'})\Sigma''_Z + \frac{1}{3}Z_{if}Z_{jf'}\Sigma'''_Z}{2(1 + \Sigma'_Z)^2} + \frac{Z_{if}Z_{jf'}(\Sigma''_Z)^2}{4(1 + \Sigma'_Z)^3}\right.$$
$$\left. + \frac{G_{if}G'_{jf'} + G'_{if}G_{jf'}}{s + \Sigma_{\gamma\gamma}} - \frac{G_{if}G_{jf'}(1 + \Sigma'_{\gamma\gamma})}{(s + \Sigma_{\gamma\gamma})^2} + B'_{ij}\right]_{s=s_0} + B^{S'}_{\gamma Z, ij}, \tag{15}$$

Here $X'$ denotes the derivative of $X$ with respect to $s$.

The vertex form factors and box diagrams can contain infrared (IR) divergencies from QED and (in the case of external quarks) QCD corrections. When interfacing the matrix elements with a Monte-Carlo (MC) program, these IR divergent contributions and the corresponding real emission contributions will be produced by the MC phase-space generator and showering algorithm. Thus they must be excluded from the hard matrix elements encoded in GRIFFIN.

For the vertex form factors, the IR-divergent contributions can be factorized, $Z_{if}^{\text{tot}} = R_f^i \times Z_{if}$, where $R_f^i$ ($i = \text{V}, \text{A}$) contain the QED/QCD corrections to the $f\bar{f}$ pair (see *e.g.* Ref. [42]). They are defined via the matrix elements for the decay of a vector boson into $f\bar{f}$:

$$R_f^{\text{V}}(s) \equiv \frac{\mathcal{M}_{V^* \to f\bar{f}}^{\text{QED/QCD}}}{\mathcal{M}_{V^* \to f\bar{f}}^{\text{Born}}}, \qquad R_f^{\text{A}}(s) \equiv \frac{\mathcal{M}_{A^* \to f\bar{f}}^{\text{QED/QCD}}}{\mathcal{M}_{A^* \to f\bar{f}}^{\text{Born}}}, \tag{16}$$

where $V^*$ ($A^*$) denotes a generic vector boson with invariant mass $s$ that couples to the $f\bar{f}$ fermion current with a pure vector (axial-vector) coupling, and the superscript "QED/QCD" indicates that all QED and QCD to the desired order are included. The factorization is not perfect, but the remaining non-factorizable contributions [43, 44] are IR-finite and can be incorporated into $Z_{if}$ order by order. Here and in the following, we adopt the notation that $Z_{if}$ is the IR-finite vertex form factor after the IR-divergent QED/QCD contributions have been factored off, whereas $Z_{if}^{\text{tot}}$ is the vertex form factor including all QED/QCD corrections.

The subtraction of these contributions is less straightforward for the box diagrams, which contain IR-divergent initial-final interference (IFI) terms. We here restrict ourselves to a discussion at NLO, where one encounters IR-divergent IFI terms from two sources, the $\gamma\gamma$ boxes, and the $\gamma$Z boxes. Following the CEEX MC scheme of Ref. [45], they can be removed with the following subtraction terms:

$$\gamma\gamma \text{ box:} \qquad B_{\text{VV}(1)} = B_{\text{VV}(1)}^{\text{tot}} - S_{\text{VV}}^{(0)} \frac{\alpha}{\pi} Q_f Q_{f'} \, f_{\text{IR}}(m_\gamma, t, u), \tag{17}$$

$$\gamma Z \text{ box:} \qquad B_{\gamma Z, ij(1)} = B_{\gamma Z, ij(1)}^{\text{tot}} - \frac{R_{ij}^{(0)}}{s - s_0} \frac{\alpha}{\pi} Q_f Q_{f'} \, [f_{\text{IR}}(m_\gamma, t, u) + \delta_G(s, t, u)], \tag{18}$$

$$f_{\text{IR}}(m_\gamma, t, u) = \frac{2\pi}{\alpha} \big[ R_{e(1)}(t) - R_{e(1)}(u) \big] = \ln\Big(\frac{1 - c_\theta}{1 + c_\theta}\Big) \bigg[ \ln\Big(\frac{2m_\gamma^2}{s\sqrt{1 - c_\theta^2}}\Big) + \frac{1}{2} \bigg],$$

$$\delta_G(s, t, u) = -2\ln\Big(\frac{1 - c_\theta}{1 + c_\theta}\Big) \ln\Big(\frac{s_0 - s}{s_0}\Big). \tag{19}$$

Here the subscripts $(n)$ indicate the loop order. $R_{e(1)}$ is the radiative factor defined in eq. (16) with one-loop QED corrections (the upper indices for denoting vector/axial vector are suppressed here since the QED correction is chiral-blind). When matching our IR-subtracted results to an MC generator, the $R$ factors can be implemented with any IR regularization scheme in the MC program since they are based on a physical process and thus scheme independent. Only for illustration, we show their form when using a small photon mass $m_\gamma$ as a regulator. The current version of GRIFFIN uses this subtraction

scheme, but other schemes for removing the IR-divergent IFI contributions could also be easily implemented.

Near the Z resonance, when aiming for a description at N$^n$LO precision, it is typically sufficient to compute only the leading coefficient $R$ to $n$-loop order, whereas $(n-1)$-loop and $(n-2)$-loop precision is adequate for $S$ and $S'$, respectively[4].

Furthermore, the ratio $\Gamma_Z/M_Z = \mathcal{O}(\alpha)$, where $\mathcal{O}(\alpha)$ denotes electroweak NLO corrections, which implies that one can perform expansions in the perturbative order, $\alpha$, and $\Gamma_Z/M_Z$ in parallel. For example, $f(s_0) = f(M_Z^2) - iM_Z\Gamma_Z f'(M_Z^2) - \frac{M_Z^2\Gamma_Z^2}{2} f''(M_Z^2) + ...$ Thus, in summary, we adopt the power counting $(s-s_0)/M_Z^2 \sim \Gamma_Z/M_Z \sim \alpha$ for the expansion of the matrix element near the Z pole.

If we wish to expand up to NNLO for the leading pole term, one would in principle, also need the $\gamma Z$ box to two-loop order, which is currently unknown. However, it was shown in Refs. [46–48] that at NLO the total contribution of IFI terms to $R_{ij}$ vanishes when adding up the virtual $\gamma Z$ boxes and real photon radiation (see also Refs. [49, 50]). This argument, of course, only holds for sufficiently inclusive observables. Furthermore, for quarks in either the initial or final state, Ref. [51] demonstrated that the resonance pole also cancels in the IFI contributions for mixed electroweak-QCD NNLO corrections. A similar argument should apply to the $\gamma\gamma Z$ boxes at electroweak NNLO, although a more careful analysis of this issue would be desirable. Assuming that this argument holds, one only needs to include $B^R_{\gamma Z(m)}$, $m = 1, ..., n-1$ for the computation of $R_{ij}^{(n)}$.

Based on the above considerations, the result for an expansion up to NNLO for the leading pole term $R$ (which implies NLO precision for $S$ and LO for $S'$) reads

$$R_{ij}^{(0)} = Z_{if(0)}Z_{jf'(0)}, \tag{20}$$

$$R_{ij}^{(1)} = \left[Z_{if(0)}Z_{jf'(1)} + Z_{if(1)}Z_{jf'(0)} - Z_{if(0)}Z_{jf'(0)}\Sigma'_{Z(1)}\right]_{s=M_Z^2}, \tag{21}$$

$$R_{ij}^{(2)} = \left[Z_{if(0)}Z_{jf'(2)} + Z_{if(2)}Z_{jf'(0)} + Z_{if(1)}Z_{jf'(1)} - Z_{if(0)}Z_{jf'(0)}\Sigma'_{Z(2)} - \Sigma'_{Z(1)}R_{ij}^{(1)}\right.$$
$$\left. - iM_Z\Gamma_Z(Z_{if(0)}Z'_{jf'(1)} + Z'_{if(1)}Z_{jf'(0)} - Z_{if(0)}Z_{jf'(0)}\Sigma''_{Z(1)})\right]_{s=M_Z^2} + B^R_{\gamma Z,ij(1)}, \tag{22}$$

$$S_{ij}^{(0)} = \frac{1}{M_Z^2}G_{if(0)}G_{jf'(0)}, \tag{23}$$

$$S_{ij}^{(1)} = \left[Z_{if(0)}Z'_{jf'(1)} + Z'_{if(1)}Z_{jf'(0)} - \frac{1}{2}Z_{if(0)}Z_{jf'(0)}\Sigma''_{Z(1)} + \frac{1}{M_Z^2}\left(G_{if(0)}G_{jf'(1)} + G_{if(1)}G_{jf'(0)}\right)\right.$$
$$\left. + \frac{iM_Z\Gamma_Z - \Sigma_{\gamma\gamma(1)}}{M_Z^4}G_{if(0)}G_{jf'(0)} + B_{ij(1)}\right]_{s=M_Z^2} + B^S_{\gamma Z,ij(1)}, \tag{24}$$

$$S_{ij}'^{(0)} = -\frac{1}{M_Z^4}G_{if(0)}G_{jf'(0)}, \tag{25}$$

where the subscripts $(n)$ again indicate the loop order.

---

[4]This power counting can be extended to more terms, beyond $S'$, in the Laurent expansion.

As mentioned in the introduction, electroweak pseudo-observables (EWPOs) are used as an intermediate step when comparing experimental data to theory expectations. The EWPOs can be expressed in terms of the form factors $F_{V,A}^f$ defined in Ref. [52][5] and in terms of the effective weak mixing angle $\sin^2 \theta_{\text{eff}}^f$ (as defined, *e.g.*, in Ref. [54]). Up to NNLO, and using the power counting $\alpha \sim \Gamma_Z/M_Z$, they are given by

$$\sin^2 \theta_{\text{eff}}^f = \frac{1}{4|Q_f|} \left[ 1 - \text{Re}\, \frac{Z_{Vf}}{Z_{Af}} \right]_{s=M_Z^2}, \tag{26}$$

$$F_A^f = \left[ \frac{|Z_{Af}|^2}{1 + \text{Re}\,\Sigma_Z'} - \frac{1}{2} M_Z \Gamma_Z |a_{f(0)}^Z|^2 \, \text{Im}\, \Sigma_Z'' \right]_{s=M_Z^2} + \mathcal{O}(\alpha^3), \tag{27}$$

$$F_V^f = \left[ \frac{|Z_{Vf}|^2}{1 + \text{Re}\,\Sigma_Z'} - \frac{1}{2} M_Z \Gamma_Z |v_{f(0)}^Z|^2 \, \text{Im}\, \Sigma_Z'' \right]_{s=M_Z^2} + \mathcal{O}(\alpha^3) \tag{28}$$

$$= F_A^f \left[ (1 - 4|Q_f| \sin^2 \theta_{\text{eff}}^f)^2 + \left( \text{Im}\, \frac{Z_{Vf}}{Z_{Af}} \right)^2 \right] \tag{29}$$

For $f = \nu$ the effective weak mixing angle is ill-defined and irrelevant, and only $F_A^\nu$ is needed.

The matrix elements for the process $f\bar{f} \to f'\bar{f}'$ can be expressed in terms of these form factors. In fact, they only enter the leading pole coefficient, $R$, as follows:

$$
\begin{aligned}
R_{ij}^{(0+1+2)} = 4 I_f^3 I_{f'}^3 \sqrt{F_A^f F_A^{f'}} &\left[ \tilde{Q}_i^f \tilde{Q}_j^{f'} \left( 1 + i\, r_{AA}^I - \tfrac{1}{2}(r_{AA}^I)^2 + \tfrac{1}{2}\delta\overline{X}_{(2)} \right) \right. \\
&\left. + (\tilde{Q}_i^f I_{j,f'} + \tilde{Q}_j^{f'} I_{i,f})(i - r_{AA}^I) - I_{i,f} I_{j,f'} \right] \\
&+ M_Z \Gamma_Z \, Z_{if(0)} Z_{jf'(0)}' \, x_{ij}^I,
\end{aligned}
\tag{30}
$$

where

$$\tilde{Q}_V^f = 1 - 4|Q_f| \sin^2 \theta_{\text{eff}}^f, \qquad\qquad\qquad \tilde{Q}_A^f = 1, \tag{31}$$

$$I_{V,f} = \frac{1}{(a_{f(0)}^Z)^2} \left[ a_{f(0)}^Z \, \text{Im}\, Z_{Vf(1)} - v_{f(0)}^Z \, \text{Im}\, Z_{Af(1)} \right], \qquad I_{A,f} = 0, \tag{32}$$

$$\delta\overline{X}_{(2)} = -(\text{Im}\, \Sigma_{Z(1)}')^2 + 2\, \frac{B_{\gamma Z, ij(1)}^R}{R_{ij}^{(0)}}, \tag{33}$$

$$r_{AA}^I = \frac{\text{Im}\, Z_{Af(1)}}{a_{f(0)}^Z} + \frac{\text{Im}\, Z_{Af'(1)}}{a_{f'(0)}^Z} - \text{Im}\, \Sigma_{Z(1)}', \tag{34}$$

$$x_{ij}^I = \frac{\text{Im}\, Z_{if(1)}'}{Z_{if(0)}} + \frac{\text{Im}\, Z_{jf'(1)}'}{Z_{jf'(0)}} - \frac{1}{2} \text{Im}\, \Sigma_{Z(1)}''. \tag{35}$$

## 3 Combination of on- and off-resonance fermion-pair production

The Laurent series (7) is only a good approximation in a window of a few GeV about the Z resonance. For values of $\sqrt{s}$ outside of this window, a non-expanded version of the matrix

---

[5]$F_A^f$ is related to $\rho_f$ introduced in Ref. [53], up to a normalization factor.

element provides a more accurate description. A unified formulation that works for a wide range of center-of-mass energies near and far from the Z resonance is given by the following prescription:

$$M_{ij} = M_{ij}^{\text{exp},s_0} + M_{ij}^{\text{noexp}} - M_{ij}^{\text{exp},M_Z^2}, \tag{36}$$

where $M_{ij}^{\text{exp},s_0}$ is the matrix element expanded about the complex pole $s_0$ as in (7), and $M_{ij}^{\text{noexp}}$ is the matrix element without any expansion in $s$ and Dyson summation. In other words, it is a straightforward fixed-order matrix element for which the full NLO electroweak corrections are known (see *e.g.* Ref. [55,56]). To avoid double counting, the expanded version of the latter, $M_{ij}^{\text{exp},M_Z^2}$ must be subtracted. Since $M_{ij}^{\text{noexp}}$ has a pole at $s = M_Z^2$, the expansion for $M_{ij}^{\text{exp},M_Z^2}$ must be performed about that point[6]:

$$M_{ij}^{\text{exp},M_Z^2} = \frac{\overline{R}'_{ij}}{(s - M_Z^2)^2} + \frac{\overline{R}_{ij}}{s - M_Z^2} + \overline{S}_{ij} + (s - M_Z^2)\overline{S}'_{ij} + \ldots \tag{37}$$

Up to NLO, the coefficients are given by

$$\overline{R}'^{(0)}_{ij} = 0, \qquad \overline{R}'^{(1)}_{ij} = -Z_{if(0)}Z_{jf'(0)}\,\Sigma_{Z(1)}\big|_{s=M_Z^2}, \tag{38}$$

$$\overline{R}^{(0)}_{ij} = Z_{if(0)}Z_{jf'(0)}, \tag{39}$$

$$\overline{R}^{(1)}_{ij} = \left[Z_{if(0)}Z_{jf'(1)} + Z_{if(1)}Z_{jf'(0)} - Z_{if(0)}Z_{jf'(0)}\Sigma'_{Z(1)}\right]_{s=M_Z^2} + B^{\overline{R}}_{\gamma Z,ij(1)}, \tag{40}$$

$$\overline{S}^{(0)}_{ij} = \frac{1}{M_Z^2}G_{if(0)}G_{jf'(0)}, \tag{41}$$

$$\begin{aligned}\overline{S}^{(1)}_{ij} = \Big[ & Z_{if(0)}Z'_{jf'(1)} + Z'_{if(1)}Z_{jf'(0)} - \frac{1}{2}Z_{if(0)}Z_{jf'(0)}\Sigma''_{Z(1)} + \frac{1}{M_Z^2}\left(G_{if(0)}G_{jf'(1)} + G_{if(1)}G_{jf'(0)}\right) \\ & - \frac{\Sigma_{\gamma\gamma(1)}}{M_Z^4}G_{if(0)}G_{jf'(0)} + B_{ij(1)}\Big]_{s=M_Z^2} + B^{\overline{S}}_{\gamma Z,ij(1)}, \end{aligned} \tag{42}$$

$$\overline{S}'^{(0)}_{ij} = -\frac{1}{M_Z^4}G_{if(0)}G_{jf'(0)}, \tag{43}$$

Note the presence of the double-pole term $\overline{R}'$, which is purely imaginary and does not exist for the complex-pole expansion. Similar to eq. (12), the $\gamma$Z box diagrams also contribute to

---

[6]$M_{ij}^{\text{exp},s_0}$ contains expansion terms $(s - s_0)^i$ for $i \leq 1$, whereas an expansion of $M_{ij}^{\text{noexp}} - M_{ij}^{\text{exp},M_Z^2}$ would have terms $(s - M_Z^2)^i$ with $i > 1$. Since the two series have different expansion points, the match between them is not perfect, but the mismatch is of order $\mathcal{O}(\Gamma_Z^2/M_Z^2)$ or $\mathcal{O}(\alpha\Gamma_Z/M_Z)$, which is beyond the level of accuracy of our results.

We also want to point out that our matching scheme for combining resonant and off-resonant regions is not unique. One could, for instance, use the complex-mass scheme [57,58] to calculate the off-resonant matrix elements, but a more careful investigation of this would be needed. At NLO our prescriptions is equivalent to the one in Ref. [20].

the single-pole term $\overline{R}$,

$$B_{\gamma Z,ij}(s,t) = \frac{B_{\gamma Z,ij}^{\overline{R}}}{s - M_Z^2} + B_{\gamma Z,ij}^{\overline{S}} + ...\tag{44}$$

The difference $M_{ij}^{\mathrm{noexp}} - M_{ij}^{\mathrm{exp},M_Z^2}$ is free of any poles at $s = M_Z^2$ and in fact it vanishes in the limit $s \to M_Z^2$. All three terms in eq. (36) are separately finite and gauge-invariant. With currently available results, $M_{ij}^{\mathrm{exp},s_0}$ can be evaluated to NNLO order near the Z pole, as described in the previous section. The current state of the art for $M_{ij}^{\mathrm{noexp}}$ is NLO, so that, for consistency, $M_{ij}^{\mathrm{exp},M_Z^2}$ should also be computed to NLO.

Both the matrix element coefficients (30) and (23)–(25), as well as the complete matrix element (36) are implemented in the GRIFFIN library.

# 4  Structure of the GRIFFIN library

The GRIFFIN package provides a framework for a hierarchy of C++ classes to compute in principle, any electroweak observable or pseudo-observable within a given model. The current version implements SM predictions for EWPOs and matrix elements for the process $f\bar{f} \to f'\bar{f}'$, with $f \neq f'$. Still, it is straightforward to include other items as well, including but not limited to:

- matrix elements for $f\bar{f} \to f\bar{f}$, with the same fermion type in the initial and final state, which includes Bhabha scattering;

- matrix elements for radiation of additional photons ($f\bar{f} \to f'\bar{f}'\gamma$) and/or fermion pairs ($f\bar{f} \to f'\bar{f}'f''\bar{f}''$), with appropriate subtraction of IR singularities (which is broadly equivalent to the concept of "electroweak pseudo-parameters" (EWPP) in section C.3 of Ref. [59]);

- predictions for EWPOs in BSM theories or in terms of effective theory extensions of the SM with higher-dimensional operators;

The library contains two base classes:

- class `inval`, which contains user-provided input parameters for a given model (such as the SM or some extension thereof);

- class `psobs`, which returns a numerical prediction for an observable or pseudo-observable, for the input parameters provided by an `inval` object.

In its basic form, `inval` simply has some basic methods for setting and retrieving the values of some input parameters. However, one can define extended classes derived from `inval` to perform computations of input parameters, such as translating between masses in the complex-pole scheme and the running-width scheme, see eq. (6), or computing the W-boson mass from the Fermi constant [60].

| Boson masses and widths | | Fermion masses | | | Couplings |
|---|---|---|---|---|---|
| $M_{\mathrm{W}}$ | $\Gamma_{\mathrm{W}}$ | $m_{\mathrm{e}}^{\mathrm{OS}}$ | $m_{\mathrm{d}}^{\overline{\mathrm{MS}}}(M_{\mathrm{Z}})$ | $m_{\mathrm{u}}^{\overline{\mathrm{MS}}}(M_{\mathrm{Z}})$ | $\alpha(0)$ |
| $M_{\mathrm{Z}}$ | $\Gamma_{\mathrm{Z}}$ | $m_{\mu}^{\mathrm{OS}}$ | $m_{\mathrm{s}}^{\overline{\mathrm{MS}}}(M_{\mathrm{Z}})$ | $m_{\mathrm{c}}^{\overline{\mathrm{MS}}}(M_{\mathrm{Z}})$ | $\Delta\alpha \equiv 1 - \alpha(0)/\alpha(M_{\mathrm{Z}}^2)$ |
| $M_{\mathrm{H}}$ | | $m_{\tau}^{\mathrm{OS}}$ | $m_{\mathrm{b}}^{\overline{\mathrm{MS}}}(M_{\mathrm{Z}})$ | $m_{\mathrm{t}}^{\mathrm{OS}}$ | $\alpha_{\mathrm{s}}^{\overline{\mathrm{MS}}}(M_{\mathrm{Z}})$ |
| | | | | | $G_\mu$ |

**Table 1:** *SM input parameters used in GRIFFIN. Here OS and $\overline{\mathrm{MS}}$ refer to the on-shell and $\overline{\mathrm{MS}}$ scheme, respectively. For most quantities currently encoded in GRIFFIN, fermion masses besides the top-quark mass are being ignored. The CKM matrix is taken to be the unit matrix. $\alpha(0)$ refers to the electromagnetic coupling in the Thomson limit, and $G_\mu$ is the Fermi constant of muon decay.*

The base version of GRIFFIN defines a set of input parameters for SM calculations, listed in Tab. 1. Most of these parameters are defined within the on-shell (OS) renormalization scheme, with the exception of light quark masses and the strong coupling, for which the $\overline{\mathrm{MS}}$ scheme is assumed (at the scale $\mu = M_{\mathrm{Z}}$). Additional input parameters for flavor physics or BSM scenarios can be easily added.

The user has the option to choose between input classes that either use $\alpha(0), M_{\mathrm{W}}, M_{\mathrm{Z}}$ or $\alpha(0), G_\mu, M_{\mathrm{Z}}$ as inputs to define the electroweak couplings. Here $\alpha(0)$ is the electromagnetic coupling in the Thomson limit, and $G_\mu$ is the Fermi constant of muon decay. An additional input is the shift $\Delta\alpha$ between the running electromagnetic couplings at the scales $q^2 = 0$ and $q^2 = M_{\mathrm{Z}}^2$. $\Delta\alpha$ receives contributions from leptons, which has been computed to four-loop order [61], and from quarks or hadrons, which can be extracted from data [62–64].

A child class descending from `psobs` can in principle encode predictions for any observable or pseudo-observable within any given model. The base version of GRIFFIN includes SM predictions for form factors, such as $\sin^2\theta_{\mathrm{eff}}^f$ and $F_{V,A}^f$, and for matrix elements for the process $f\bar{f} \to f'\bar{f}'$ near the Z resonance, using the complex pole expansion described in the previous section.

GRIFFIN version 1.0 contains the following SM corrections:

- Complete one-loop corrections for $\sin^2\theta_{\mathrm{eff}}^f$ [53,65] are implemented in the class `SW_SMNLO`. On top of this, electroweak [66–71] and mixed electroweak-QCD [72–76] two-loop corrections, as well as partial higher-order corrections are available in the class `SW_SMNNLO`. The latter include $\mathcal{O}(\alpha_t\alpha_s^2)$ [77,78], $\mathcal{O}(\alpha_t^2\alpha_s)$, $\mathcal{O}(\alpha_t^3)$ [79,80] and $\mathcal{O}(\alpha_t\alpha_s^3)$ [81–83] corrections in the limit of a large top Yukawa coupling $y_{\mathrm{t}}$, where $\alpha_t \equiv y_{\mathrm{t}}^2/(4\pi)$, and leading fermionic three-loop corrections of orders $\mathcal{O}(\alpha^3)$ and $\mathcal{O}(\alpha^2\alpha_s)$ [84, 85]. In addition, non-factorizable $\mathcal{O}(\alpha\alpha_s)$ $Zq\bar{q}$ vertex contributions [43,86–90] are also implemented in `SW_SMNNLO`.

- Similarly, the classes `FA_SMNLO` and `FV_SMNLO` provide one-loop corrections [53] for the form factors $F_{V,A}^f$, whereas `FA_SMNNLO` and `FV_SMNNLO` contain electroweak [52,91–93]

and mixed electroweak-QCD [72–76] two-loop corrections, as well as the partial higher-order corrections and non-factorizable contributions mentioned in the previous bullet point.

- For the process $f\bar{f} \rightarrow f'\bar{f}'$: The class `mat_SMNNLO` computes the matrix element according to section 3 with the following ingredients:

  – All contributions needed to compute the matrix element coefficient $R$ to NNLO accuracy according to (30), and the coefficients $S$ and $S'$ to NLO and LO, respectively, see eqs. (23)–(25). These are also separately available in the member functions `coeffR`, `coeffS`, `coeffSp` of `mat_SMNNLO`.

  – The off-resonance contribution, $M_{ij}^{\mathrm{noexp}} - M_{ij}^{\mathrm{exp}, M_Z^2}$, to NLO precision, see section 3, which is also separately available via the member function `resoffZ`.

- When using the input parameter set $\alpha(0), G_\mu, M_Z$, one needs to compute $M_W$ from these quantities according to

$$G_\mu = \frac{\pi\alpha}{\sqrt{2}M_W^2(1 - M_W^2/M_Z^2)}(1 + \Delta r). \tag{45}$$

Here $\Delta r$ accounts for radiative corrections. The class `dr_SMNNLO` contains all higher-order corrections discussed in Ref. [60], plus the leading fermionic three-loop corrections of orders $\mathcal{O}(\alpha^3)$ and $\mathcal{O}(\alpha^2\alpha_s)$ [84, 85]. These corrections are used in the input classes `invalGmu` and `SMvalGmu`.

For any of these quantities, QED and QCD corrections on the external legs have been factored out, as explained in detail in section 2. The logic is that QED/QCD effects depend on detector acceptance and selection cuts and are best simulated with MC methods. GRIFFIN could be interfaced with suitable MC tools to provide the hard electroweak matrix elements.

# 5 Sample results and comparisons

In this section, we show numerical comparisons between GRIFFIN and the DIZET library of EW radiative corrections [14, 15, 94] for the EWPOs and the differential cross-section. For the latter, we use some of the computational frameworks of the KKMCee project[7] [95].

We first perform a benchmark test of the EWPOs in comparison with DIZET v 6.45 [94], including NNLO and leading NNNLO corrections. In DIZET, the form factor is defined as in Eq. 2.4.9 and Eq. 2.4.10 of Ref. [14].

$$\Gamma_{Z \rightarrow f\bar{f}} = \Gamma_0 c_f |\rho_Z^f| (|g_Z^f|^2 R_V^f + R_A^f) + \delta_{\alpha\alpha_s}, \tag{46}$$

where we have neglected all lepton masses, $c_f = N_c^f$ is the number of colors, and

$$\Gamma_0 = \frac{G_\mu M_Z^3}{24\sqrt{2}\pi}. \tag{47}$$

---

[7]The authors are grateful to S. Jadach for sharing a suitable test program with us.

In eq. (46), $g_Z^f$ is a complex-valued variables, which in our notation from section 2 is given by

$$g_Z^f = \frac{Z_{Vf}}{Z_{Af}}, \tag{48}$$

On the other hand, in GRIFFIN, we define the partial width of Z-boson as

$$\Gamma_{Z \to f\bar{f}} = \frac{N_c^f M_Z}{12\pi}(F_V^f R_V^f + F_A^f R_A^f). \tag{49}$$

By setting this equal to eq. (46) we obtain the relation

$$\frac{N_c^f M_Z}{12\pi} F_A^f \left(\frac{F_V^f}{F_A^f} R_V^f + R_A^f\right) = \frac{N_c^f M_Z}{12\pi} F_A^f (|g_Z^f|^2 R_V^f + R_A^f) = \Gamma_0 c_f |\rho_Z^f|(|g_Z^f|^2 R_V^f + R_A^f), \tag{50}$$

This implies the following relation between $|\rho_Z^f|$ and $F_A^f$:

$$|\rho_Z^f| = \frac{2\sqrt{2}F_A^f}{G_\mu M_Z^2} \tag{51}$$

One should notice that the non-factorizable mixed QCD-EW corrections are considered as an additive part to the Z widths in DIZET, whereas in GRIFFIN, they are absorbed in the form factors $F_{V,A}$. This will cause a small numerical mismatch when comparing the form factor $F_A$ to $\rho_Z^f$. Besides, one also has to notice that eq. 51 is the modulus of $\rho_Z^f$ instead of Re $\rho_Z^f$. Hence to compare these observables, we need to use both the Im $\rho_Z^f$ and Re $\rho_Z^f$ output from DIZET to reconstruct $|\rho_Z^f|$.

The flags used by DIZET V.6.45 are listed as follows:

| | | | | |
|---|---|---|---|---|
| IHVP=5 | IAMT4=8 | IQCD=3 | IMOMS=1 | IMASS=0 |
| ISCRE=0 | IALEM=0 | IMASK=0 | ISCAL=0 | IBARB=2 |
| IFTJR=1 | IFACR=2 | IFACT=0 | IHIGS=0 | IAFMT=3 |
| IEWLC=0 | ICZAK=1 | IHIGS=1 | IALE2=3 | IGFER=2 |
| IDDZZ=1 | IAMW2=1 | ISFSR=1 | IDMWW=0 | IDSWW=0 |

Due to the limited options of EW input schemes offered by subroutines in DIZET, we have to set $G_\mu$, $M_Z$ as inputs and use the DIZET outputs for $M_W$ and $\Gamma_{W,Z}$ as inputs for GRIFFIN, as shown in the following table:

| GRIFFIN input parameters | |
|---|---|
| DIZET input parameters | DIZET output |
| $\alpha_s(M_Z^2) = 0.118, \quad \alpha = 1/137.035999084$ | $\Gamma_Z = 2.495890$ GeV |
| $\Delta\alpha = 0.059, \quad M_Z = 91.1876$ GeV, $\quad G_\mu = 1.166137 \times 10^{-5}$ | $M_W = 80.3599$ GeV |
| $m_t = 173.0$ GeV, $\quad M_H = 125.0$ GeV, $\quad m_{e,\mu,\tau,u,d,s,c,b} = 0$ GeV | $\Gamma_W = 2.090095$ GeV |

| | DIZET 6.45 | GRIFFIN all orders | GRIFFIN $\mathcal{O}(\alpha, \alpha^2, \alpha_t\alpha_s, \alpha_t\alpha_s^2)$ |
|---|---|---|---|
| $\Delta r$ | $3.63947 \times 10^{-2}$ | $3.68836 \times 10^{-2}$ | $3.63987 \times 10^{-2}$ |

| | $\lvert\rho_Z^f\rvert$ | | $\sin^2\theta_{\text{eff}}^f$ | | $\Gamma_{Z\to f\bar{f}}$ | |
|---|---|---|---|---|---|---|
| | DIZET 6.45 | GRIFFIN | DIZET 6.45 | GRIFFIN | DIZET 6.45 | GRIFFIN |
| $\nu\bar{\nu}$ | 1.00800 | 1.00814 | 0.231119 | NAN | 0.167206 | 0.167197 |
| $\ell\bar{\ell}$ | 1.00510 | 1.00519 | 0.231500 | 0.231534 | 0.083986 | 0.083975 |
| $u\bar{u}$ | 1.00578 | 1.00573 | 0.231393 | 0.231420 | 0.299938 | 0.299958 |
| $d\bar{d}$ | 1.00675 | 1.00651 | 0.231266 | 0.231309 | 0.382877 | 0.382846 |
| $b\bar{b}$ | 0.99692 | 0.99420 | 0.232737 | 0.23292 | 0.376853 | 0.377432 |

**Table 2:** *The numerical comparison of the EWPOs and form factors $\rho$ between DIZET and GRIFFIN. The partial width results are for a single fermion family. See text for details.*

With these inputs, we find the numerical results for the form factors $\Delta r$, $\lvert\rho_Z^f\rvert$ and $\sin^2\theta_{eff}^f$, as well as the partial Z width $\Gamma_{Z\to f\bar{f}}$ shown in Tab. 2. Both the DIZET and GRIFFIN results for the form factors include full $\mathcal{O}(\alpha^2)$ corrections, $\mathcal{O}(\alpha\alpha_s)$ and $\mathcal{O}(\alpha\alpha_s^2)$ QCD corrections, and leading higher-order corrections in an expansion in $m_t^2$ of $\mathcal{O}(\alpha_t^3)$, $\mathcal{O}(\alpha_t^2\alpha_s)$, and $\mathcal{O}(\alpha_t\alpha_s^3)$, but with the following differences: (a) the GRIFFIN result for $\lvert\rho_Z^f\rvert$ additionally includes non-factorizable EW-QCD corrections; (b) the DIZET result for $\Delta r$ does not include the $\mathcal{O}(\alpha_t\alpha_s^3)$ contributions, and the $\mathcal{O}(\alpha\alpha_s, \alpha\alpha_s^2)$ terms are computed only in a large-$m_t$ approximation.

A better agreement for $\Delta r$ is obtained when adjusting GRIFFIN to match the order of $\Delta r$ in DIZET, by only summing corrections of $\mathcal{O}(\alpha, \alpha^2, \alpha_t\alpha_s, \alpha_t\alpha_s^2)$. In this case, one finds a 4-digit agreement, as shown in Tab. 2.

Most predictions given by both programs for $\lvert\rho_Z^f\rvert$, $\sin^2\theta_{\text{eff}}^f$ and $\Gamma_{Z\to f\bar{f}}$ agree with each other by at least four decimal points. As aforementioned, the definition of the effective weak-mixing angle at $f = \nu$ is ill-defined. Owing to an alternative definition of $\sin^2\theta_{\text{eff}}^f$ by DIZET (see eq. 5.6 in Ref. [15]), a number is yet produced without phenomenological implications. The discrepancy is mildly larger for the form factors $\lvert\rho_Z^d\rvert$ and $\lvert\rho_Z^b\rvert$ for quark final states, which reflects the different implementations of the non-factorizable EW-QCD corrections (as mentioned above, these are encapsulated in the form factors in GRIFFIN, but treated separately in DIZET). This is especially true for $\lvert\rho_Z^b\rvert$, where the top quark comes into play for these types of corrections. However, these implementation differences do not affect the predictions for the partial widths at the given order, and indeed one can see from the table the numbers for $\Gamma_{Z\to q\bar{q}}$ agree better.

Let us now move on to comparisons of predictions for the differential cross-section, where for concreteness, we focus on the process $e^+e^- \to \mu^+\mu^-$. Within GRIFFIN, these predictions have been computed using the class `mat_SMNNLO`, whereas for `Dizet 6.45`, they are based on outputs of the subroutine `ROKANC`, which have been assembled into predictions for the differential cross-section using the KKMCee framework. Given that the $\gamma Z$ box contribution is not included in DIZET, we also turned off the $\gamma Z$ contributions in GRIFFIN for this

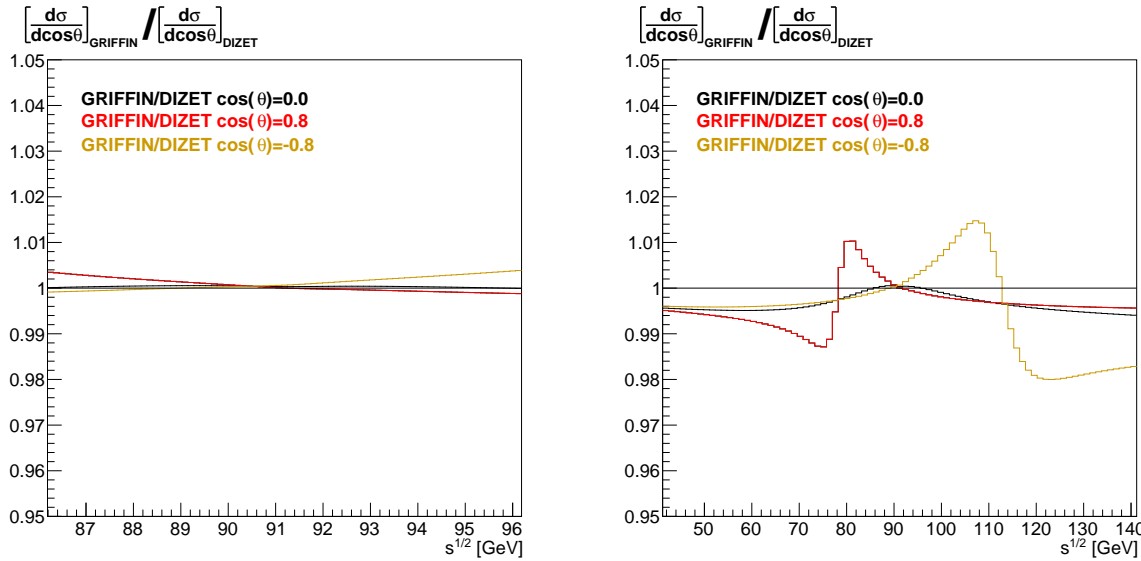

**Figure 1:** *Ratios of the differential cross-section for $e^+e^- \to \mu^+\mu^-$ using predictions from GRIFFIN v1.0 and DIZET 6.45, for three choices of the scattering angle $\theta$. The left plot is focused on the Z-pole region, while the right plot shows a wider range of center-of-mass energies.*

comparison for consistency. The results are shown in Fig. 1.

The left plot in the figure shows that there is very good agreement between GRIFFIN v1.0 and DIZET 6.45 in the Z-pole region, with deviations of $\mathcal{O}(10^{-3})$ or less. When analyzing a larger range of center-of-mass energies, as shown in the right plot, one finds larger discrepancies at the level of 0.5–2%. This is not surprising since away from the Z resonances, both codes only deliver NLO precision, and differences in the implementation in GRIFFIN and DIZET would be of NNLO. In particular, DIZET does not use the manifestly gauge-invariant pole expansion scheme described in sections 2 and 3. Note that the relative corrections in some kinematic regions (*e.g.* $\cos\theta = +0.8$ below the resonance and $\cos\theta = -0.8$ above the resonance) are enhanced due to strong cancellations between the s-channel photon and Z exchange contributions, which render the tree-level matrix element small. In these regions, the NLO corrections reach 20–30%, so that $\mathcal{O}(\%)$ discrepancies from missing NNLO contributions are perfectly consistent with expectations.

We also wish to note that GRIFFIN provides a framework that can be systematically extended to higher orders, and NNLO corrections for $e^+e^- \to f\bar{f}$ can be included in the library once they become available. With this improvement, the theory uncertainty for the differential cross-section will likely be reduced significantly below 1%.

# 6 Summary

The GRIFFIN library provides a consistent description of the IR-subtracted matrix elements for fermion scattering for a wide range of center-of-mass energies, including close to and far away from the Z-boson resonance. This is achieved by merging a complex-pole expansion, which provides an accurate description near the Z peak, with an unexpanded fixed-order calculation, which is more adequate outside of the Z resonance region.

Version 1.0 of the library includes all currently available higher-order SM corrections for the leading Z-pole term, and NLO SM corrections for the remainder. It also includes two input parameter schemes, which either use the Fermi constant $G_\mu$ or the W mass $M_W$ as inputs and all available higher-order SM corrections for the translation between the two. The results have been validated and compared to the DIZET 6.45 library.

The structure of GRIFFIN is modular and object-oriented to easily facilitate future extensions. Possible such extensions are:

- Higher-order corrections for the Z-pole form factors and matrix elements;

- Matrix elements for Bhabha scattering and/or final states with additional partons;

- Predictions for BSM scenarios, including effective theory frameworks such as SMEFT;

- Implementation of different schemes for factorization of initial- and final-state radiation;

- Implementation of other processes, such as charged-current Drell-Yan production with a W-boson resonance, or W-boson decays.

The authors invite the community to contact them with feedback, suggestions for improvement, or to contribute new modules to include in the GRIFFIN system.

## Acknowledgments

The authors are indebted to S. Jadach and J. Holeczek for providing us with a test program within the KKMCee framework for producing comparisons, as well as for numerous discussions. We also thank Z. Wąs for helpful communications. This work has been supported in part by the U.S. National Science Foundation under grant no. PHY-2112829.

## A   Appendix: Download information

The GRIFFIN source code is available at `https://github.com/lisongc/GRIFFIN/releases`. The manual is also available at `https://github.com/lisongc/GRIFFIN_manual`.

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
