# Peer review of "GRIFFIN: A C++ library for electroweak radiative corrections in fermion scattering and decay processes"

_SciPost Physics Codebases_

## Round 1 · Referee Report · Anonymous (Referee 1) · 2023-2-7

Strengths

• The code can be systematically extended to include higher-order corrections and new-physics models.
• The code will be relevant for future lepton colliders and supersedes software packages used at LEP 1.
• The implementation has been verified against the DIZET library and includes all available higher-order corrections for the leading pole term.

Weaknesses

• The differences between GRIFFIN and DIZET in the differential cross section are not understood.
• The paper contains several typos.

Report

The paper describes a modular framework for the calculation of electroweak scattering and decay processes including perturbative corrections. It is specifically realised for processes with 4 external fermions and combines a pole expansion near a vector-boson resonance with a fixed-order calculation in the non-resonant region. Owing to its modular structure, the library can be easily extended to include further higher-order corrections or predictions in new-physics models. In its present state, the code includes all known corrections to fermion-pair production on and off the Z resonance. This is relevant for the investigation of fermion-pair production at future electron-positron colliders.

Requested changes

• The formulation in Section 2 relies on the literature for the corresponding calculations for LEP 1. The due credit to the original papers should be given.
• It should be made clear, how logarithms of the form ln(1 − s/s 0 ) are treated in the pole expansion.
• The code contains only the IR-finite part of the NLO corrections. The precise definition of this part should be given, for instance for the vertex form factors.
• The cancellation of IFI between γZ boxes and the corresponding real radiation mentioned on page 5 is only valid for inclusive quantities. This should be stressed.
• While it is obvious that Eq. (35) does not lead to double counting close to the resonance, this is not so clear away from the resonance. The authors should comment on this point.
• The “etc.” in the first list of items on page 9 should be explained or omitted.
• In the comparison between GRIFFIN and DIZET the largest differences show up for quantities related to bottom quarks. This needs to be commented.
• The differences between GRIFFIN and DIZET of up to 2% in the differential cross section away from the resonance region appear to be quite large. Can these really be explained by NNLO effects? Which enhancements of NNLO corrections can cause differences of this size?

  • validity: ok
  • significance: good
  • originality: good
  • clarity: good
  • formatting: excellent
  • grammar: good

Author:  Ayres Freitas  on 2023-04-11  [id 3574]

(in reply to Report 1 on 2023-02-07)
Category:
reply to objection

Please see attached file for our response to the referee report

Attachment:

response_letter_griffin.pdf

---

## Round 1 · Referee Report · Anonymous (Referee 2) · 2023-2-11

Strengths

  • State-of-the art precision calculations valid in the resonance region are matched to off-shell calculations in a systematic way

  • Thanks to the modular C++ implementation, this code can be extended in various directions and interfaced to a variety of Monte Carlo tools.

Weaknesses

  • The implemented factorization scheme for IR singularities does not corresponds to the conventional subtraction schemes (e.g. Catani-Seymour or FKS) that are used in modern Monte Carlo tools for high-energy colliders.

Report

I believe that this code meets the requirements for publication in SciPost Physics Codebases. Its content is well documented, and its modular and flexible structure opens the door to a variety of applications at high-energy colliders.

Requested changes

1) The title, abstract and introduction give the impression that GRIFFIN may be applicable to cross sections, while it only computes IR-subtracted $2\to 2$ matrix elements. This should be stated in a more explicit way, starting from the abstract.

2) The treatment of logarithms of $1-s/s_0$ in GRIFFIN should be described in some details.

3) The implemented IR-factorization prescription should be documented in full detail, i.e. with explicit formulas for all factorized singularities. This information is crucial for interfacing the code to Monte Carlo generators.

4) The IR factorisation scheme should also be adapted to dimensional regularisation (with massless photons and massive or massless fermions)

5) The matching prescription (35) involves off-shell matrix elements with real Z-boson masses. The authors should comment on its applicability to modern off-shell calculations based on the complex-mass scheme.

6) The terminology "cross section matrix elements" at the beginning of Sect. 4 is confusing and should be clarified.

7) Table 1: the authors should clearly indicate which parameters play the role of user-provided input parameters and which ones are derived from other input parameters. For instance, gauge-boson widths are not independent input parameters: how are they treated in GRIFFIN?

8) The origin of the percent-level differences in Fig. 1 should be clarfied in some detail.

  • validity: good
  • significance: ok
  • originality: ok
  • clarity: good
  • formatting: good
  • grammar: excellent

Author:  Ayres Freitas  on 2023-04-11  [id 3575]

(in reply to Report 2 on 2023-02-11)
Category:
reply to objection

Please see attached file for our response to the referee report

Attachment:

response_letter_griffin_FmHTGAH.pdf

---

## Round 1 · Referee Report · Anonymous (Referee 3) · 2023-2-13

Strengths

  • describes a valuable software package for precision calculations
  • provides enough theory overview to inform the content of the package without having to resort to the original literature
  • generally well-written

Weaknesses

  • some choices made in the implementation are not ideal for modern calculations, e.g. the choice of infrared subtraction scheme, missing complex mass schemes, etc
  • differences to Dizet not sufficiently explained

Report

This manuscript describes a new software library that has the potential to be very useful for precision calculations for some precision calculations at future e+e- colliders.

Requested changes

1) p.1, first paragraph: Measurements of DY at the LHC and the Tevatron also include initial state b-quarks, whose contributions play a role a the precision provided by the LHC experiments. Of course, their contribution at higher-order EW differs from the other light quarks due to the top, but they should at least be mentioned here. 2) p.2, first paragraph: Along the examples for QED corrections in MC for e+e- a few more recent advances should be mentioned, in particular 1911.12040 and 2203.10948, possibly refering to 2203.12557 for an overview. 3) p.3, eq. (1): the meaning of the $\otimes$ symbol should be introduced. 4) p.3, eq. (3): $c_\theta$ is undefined. 5) p.3, eq. (3): $s$ is undefined. 6) p.3, eq. (5): $s_w$ and $c_w$ are undefined. 7) p.3, eq. (6): Possibly connect eq. (6) to the otherwise well-known conversion between pole and on-shell scheme, which I think are the same just using different terminology. 8) It is not entirely clear how logarithms of $1-s/s_0$ are handled in the expansion of eq. (35) and following, please elaborate. 9) In the matching of the on-shell resummation to the complete off-shell fixed-order calculation, it is not obvious that no artifacts appear away from the on-shell limit as the resummation is never switched off. Please comment. 10) Further, also in the limit that the off-shell calculation is in the vicinity of the on-shell production, since the pole is never reached by the off-shell calculation due to the finite width, it is not clear either that no artifacts are introduced as in this limit the expanded resummation and the fixed-order do not (logarithmically) coincide (at least when a modern universal scheme like the complex mass scheme is used). 11) The details on the implementation of the IR subtraction are insuffient, in particular if the code is to be used and interfaced to other tools and event generators. In particular, with the limited information provided a conversion to modern subtraction schemes using dimensional regularisation is not straightforward. 12) In the comparison to Dizet, the observed differences are not sufficiently explained in text, in particular in processes with bottom quarks. Please elaborate. In particular since the deviations of up to 2% naively seem too large to be attributed to generic NNLO effects. If they are enhanced by some mechanism, please discuss these.

---

## Editorial Decision

resubmitted